# Edge-colored Clustering in Hypergraphs: A MaxECC Approximation

**Aravind Srinivasan** [* 1]  **Arushi Srinivasan** [* 1]  **Jiayi Wu** [* 1]

## Abstract

We study the MAXECC problem, where given an edge-colored hypergraph with $k$ colors and edge size $r$, we seek to color the vertices of the graph in order to maximize the number of satisfied edges (edges having the same color as their extremities): this is an effective mechanism for clustering (coloring) objects based on their multi-way interactions with one another in a system, providing significant applications in machine learning, clustering, and data mining. We exponentially improve upon the approximation ratio of an existing algorithm to $\frac{1}{r+1}$, present another novel dependent-rounding algorithm with an approximation ratio of $1/\lceil \frac{k}{2} \rceil$, and modify the initial algorithm via analytical scaling techniques in order to achieve an approximation factor of $(1 - e^{-r})/r$. We then apply our scaling algorithm to graph MAXECC and improve the best-known approximation factor for all hypergraphs: in particular, our algorithm provides an approximation factor of $0.43$ as opposed to the previously-known $0.38$ factor for graphs.

## 1. Introduction

Clustering is a fundamental technique in unsupervised learning. We significantly improve upon the approximation algorithms for a natural edge-colored clustering problem studied in an ICML 2025 paper of (Crane et al., 2025) and earlier (with applications, e.g., to temporal hypergraph clustering, inferring researchers' fields based on publications, assigning tasks based on prior team experiences etc.) as discussed in detail next.

Recall the definition of a *hypergraph* $H$: it contains a set $V$ of vertices and a collection $E$ of "hyperedges", where each hyperedge is simply a subset of $V$. Thus, a graph is a special case of a hypergraph in which all hyperedges have size 2.

Clustering is a fundamental problem in machine learning and data mining, with applications spanning community detection, image segmentation, and collaborative filtering (Bansal et al., 2004; Bonchi et al., 2012). Edge-colored clustering (ECC) extends traditional clustering frameworks to handle *categorical* relationships between data points. Formally, an ECC instance consists of a hypergraph $H = (V, E, \ell)$ where vertices represent data objects, hyperedges capture multi-way interactions, and an edge-coloring function $\ell : E \to [k]$ assigns each edge a color from a palette of $k$ colors representing interaction types or categories. Here and throughout, the notation "$[t]$" for a positive integer $t$ refers to the set $\{1, 2, \ldots, t\}$; so, our set of colors here is $\{1, 2, \ldots, k\}$.

The ECC framework has been successfully applied to diverse domains. In temporal hypergraph clustering (Amburg et al., 2020), edge colors encode time windows, enabling the identification of vertices that are especially active during specific periods. In team formation problems (Amburg et al., 2022), vertices represent individuals, edges correspond to team tasks, and colors indicate task types; ECC then assigns tasks based on prior collaborative experiences. In author-publication networks, vertices may be researchers, edges are co-authorship sets, and colors denote publication venues, allowing inference of researchers' primary fields.

Two natural objective functions arise in ECC: maximizing the number of *satisfied* edges (MAXECC) and minimizing the number of *unsatisfied* edges (MINECC). An edge $e \in E$ is *satisfied* under a vertex coloring $\lambda : V \to [k]$ if every vertex $v \in e$ receives the edge's color, i.e., $\lambda(v) = \ell(e)$ for all $v \in e$. While these objectives coincide at optimality, they differ significantly in efficient approximability, with MINECC being the easier of the two (Veldt, 2023; Crane et al., 2025). More generally, the focus in MAXECC is, given a non-negative weight for each edge, to maximize the total weight of the satisfied edges (with all edges having weight 1 connoting the special case of maximizing the number of satisfied edges).

---

[1]Department of Computer Science, University of Maryland, College Park, United States. Correspondence to: Aravind Srinivasan <asriniv1@umd.edu>, Arushi Srinivasan <asrin708@terpmail.umd.edu>, Jiayi Wu <jwu12328@terpmail.umd.edu>.

*Proceedings of the $43^{rd}$ International Conference on Machine Learning*, Seoul, South Korea. PMLR 306, 2026. Copyright 2026 by the author(s).

## 1.1. Prior Work

Note that graphs are a special case of hypergraphs with each hyperedge having size 2. The study of edge-colored clustering on the special case of graphs was initiated by Angel et al. (Angel et al., 2016), who established NP-hardness and provided the first approximation algorithms for MAXECC with ratio $e^{-2} \sim 0.135$;[1] recall that for $0 < \rho \le 1$, a polynomial-time "approximation algorithm" for the problem has *approximation ratio* $\rho$ if it always delivers a solution of value at least $\rho$ times optimal.[2] Subsequent work improved this guarantee through increasingly sophisticated linear program (LP)-rounding techniques (Ageev & Kononov, 2015; Alhamdan & Kononov, 2019; Ageev & Kononov, 2020), culminating in a $\frac{154}{405} \approx 0.38$ approximation for graphs (Crane et al., 2025). Please see (Angel et al., 2016) for an additional connection to an information-sharing model of (Kleinberg & Ligett, 2013).

For hypergraphs, Amburg et al. (Amburg et al., 2020) extended ECC and provided 2-approximation algorithms for MINECC. Veldt (Veldt, 2023) gave a combinatorial 2-approximation algorithm for MINECC and established "UGC-hardness" for any constant factor better than 2; since our focus is on MAXECC, we do not define the Unique-Games Conjecture (UGC) and UGC-hardness here. Given a hypergraph, its *rank* is the largest size of any hyperedge in it; the rank of any graph is two. For MAXECCon hypergraphs, Crane et al. (Crane et al., 2025) recently provided the first approximation algorithms, achieving ratio $\left(\frac{2}{e}\right)^r \cdot (r+1)^{-1}$ for rank-$r$ hypergraphs through LP-rounding. This ratio, however, diminishes exponentially fast in $r$ as the rank increases, since $2/e \sim 0.736 < 1$.

Recently, Lee et al. (Lee et al., 2025) introduced LP-based combinatorial algorithms for certain *overlapping* and *robust* variants of ECC, demonstrating that "primal-dual" approximation methods can achieve strong theoretical guarantees while maintaining computational efficiency for these extensions of ECC. These extensions are outside the scope of this work.

## 1.2. Our Contributions

We present three main contributions, which significantly improve upon the approximation ratio $\left(\frac{2}{e}\right)^r \cdot (r+1)^{-1}$ of (Crane et al., 2025).

**(1) Improved analysis of existing algorithm.** We provide

---

[1] We use $e$ both to denote the base $2.718\ldots$ of the natural logarithm as in our usage of $e^{-2}$ here, and to denote a hyperedge of $H$: there will be no confusion as to which interpretation to choose in any of our usage.

[2] Approximation algorithms are one natural way to deal with NP-hardness; for maximization problems such as MAXECC, one aims for as large a $\rho$ as possible.

*Table 1.* Comparison of approximation ratios for MAXECC

| $r$ | (Crane et al., 2025) | Ours (Analysis) | Ours (Scaling) |
|---|---|---|---|
| 2 | 0.180, 0.38 | 0.333 | **0.432** |
| 3 | 0.100 | 0.250 | **0.317** |
| 4 | 0.059 | 0.200 | **0.245** |
| 5 | 0.036 | 0.167 | **0.199** |
| 10 | 0.004 | 0.091 | **0.100** |

a refined analysis of the randomized rounding algorithm of Crane et al. (Crane et al., 2025), improving the approximation ratio from $\left(\frac{2}{e}\right)^r \cdot (r+1)^{-1}$ to $\frac{1}{r+1}$ for rank-$r$ hypergraphs. Note that our approximation only scales roughly as $1/r$, as compared to the exponential decay in $r$ as in the bound of (Crane et al., 2025). Although the algorithm remains unchanged, our analysis identifies that the previous bound was loose.

**(2) Dependent-rounding approach.** We develop a novel rounding algorithm based on *dependent rounding* (Gandhi et al., 2006; Srinivasan, 2001) that achieves a $\frac{1}{\lceil k/2 \rceil}$ approximation where $k$ is the number of colors. This approach is particularly effective when the number of colors is small relative to the hypergraph rank, and introduces dependent-rounding techniques to the ECC literature for the first time.

**(3) Scaling technique with refined probabilistic analysis.** We introduce a scaling function that modifies a key thresholding step of the randomized algorithm of (Crane et al., 2025). Through careful analysis beyond standard Jensen's inequality–specifically, by proving that certain objective functions are minimized when tentative color assignments are balanced across vertices——we establish an approximation ratio of $\frac{1-e^{-r}}{r}$, which is a further improvement of our $\frac{1}{r+1}$ above. Of particular note is the following. For the useful special case of graphs where $r = 2$, the work of (Crane et al., 2025) develops a specialized approximation algorithm with a ratio of 0.38; our $\frac{1-e^{-r}}{r}$ improves this to approximately 0.432. (Of course, we also obtain improvements for all $r$ as well.)

Table 1 compares our results with prior work across different hypergraph ranks $r$. Note in particular our significant multiplicative improvements as $r$ increases, since the term $\frac{2}{e}^r$ in the result of (Crane et al., 2025) decays to 0 exponentially fast as a function of $r$.

**Perspective.** We obtain significant algorithmic improvements over an ICML 2025 paper (and earlier works) on clustering (Crane et al., 2025); applications include several variants of clustering data that are characterized by categorical relationships among data points, as described in (Crane et al., 2025) and earlier. This is rigorous theoretical work with most proofs deferred to the Appendix; we do not conduct empirical evaluations in this version.

## 2. Preliminaries

### 2.1. Problem Definition

An instance of edge-colored clustering is given by a hypergraph $H = (V, E, \ell)$ where $V$ is a node set, $E$ is a set of hyperedges (each element of $E$ being a "(hyper)edge" that is a subset of $V$), and where $\ell : E \to [k]$ is a given mapping from the edges to a color set $[k] = \{1, 2, \ldots, k\}$. In addition, each $e \in E$ also has a given *weight* or importance, $w_e \geq 0$. We denote the *rank* of $H$ (the maximum hyperedge size) as $r = \max_{e \in E} |e|$.

The goal of ECC is to construct a map $\lambda : V \to [k]$ that associates each node with a color such that edge satisfaction is optimized. An edge $e \in E$ is *satisfied* if $\lambda(v) = \ell(e)$ for all $v \in e$, and *unsatisfied* otherwise. We denote by $Z_e$ the indicator that edge $e$ is satisfied, and every edge has a nonnegative weight $w_e$ associated with it.

In MAXECC, the **objective** is to maximize the weighted sum of satisfied edges.

### 2.2. Integer Programming and LP Formulations

MAXECC can be formulated as the following integer program:

$$\max \quad \sum_{e \in E} \omega_e \, z_e$$

$$\text{s.t.} \quad \sum_{c=1}^{k} x_v^c = 1 \quad \forall v \in V,$$

$$x_v^c \geq z_e \quad \forall c \in [k], \ \forall e \in E_c, \ \forall v \in e,$$

$$x_v^c \in \{0, 1\} \quad \forall c \in [k], \ \forall v \in V,$$

$$z_e \in \{0, 1\} \quad \forall e \in E \quad (1)$$

Setting $x_u^c = 1$ indicates that node $u$ receives color $c$ (i.e., $\lambda(u) = c$), and $z_e = 1$ indicates that edge $e$ is satisfied.

Since integer programming is $NP$-hard while linear programs (LPs) can be solved efficiently in theory and practice, we naturally consider the LP relaxation: this is obtained by relaxing the binary constraints to $x_v^c, z_e \in [0, 1]$.

A solution to the LP relaxation gives a fractional node-color assignment where $x_v^c \in [0, 1]$ can be interpreted as the extent to which vertex $v$ "wants" color $c$. The algorithmic challenge is to round these fractional values to values in $\{0, 1\}$ in order to produce an integral coloring while maintaining robust approximation guarantees. The optimal LP-relaxation value is an *upper bound* on our optimal integer program value; thus, it will suffice to show a feasible rounded solution whose expected[3] value is at least $\rho$ times

---

[3]"Expected" since we allow *randomized* rounding algorithms.

the optimal LP-solution value (for as large a $\rho$ as we can obtain); this will automatically imply a $\rho$-approximation.

We restate the following observations in (Crane et al., 2025) below as well as their original randomized ECC algorithm (Algorithm 1).

**Observation** 1. Given a randomized ECC algorithm $A$ and a fixed constant $p \in [0, 1]$, if for each $e \in E$, we have that $\Pr[e \text{ is satisfied by } A] \geq p z_e$, then $A$ is a p-approximation.

We denote $X_{v,c}$ as the event that node $v$ wants color $c$ and $Z_e$ as the event that every $v \in e$ wants color $c_e = \ell(e)$.

**Observation** 2. For every $v \in V$, $\Pr[X_{v,c}] = \Pr[\alpha_c < x_{v,c}] = x_{v,c} \leq 1$ and the events $\{X_{v,c}\}_{c \in [k]}$ are independent.

**Observation** 3. $\Pr[\cap_{v \in e} X_{v,c}] = \min_{v \in e} x_{v,c} = z_e$.

In order to ensure, for each $c \in [k]$ and $e \in E$, that the events $X_{v,c_{v \in e}}$ are positively correlated, their algorithm generates color thresholds $\alpha_c$ uniformly at random from $[0, 1]$ so that each event $X_{v,c}$ occurs if and only if $x_{v,c} < \alpha_c$. Moreover, let $\mathcal{W}_e = \{c \in [k] \mid X_{v,c}\}$ for every edge $e$.

## 3. Improved Analysis: $\frac{1}{r+1}$-Approximation

In this section, we provide an improved analysis of the randomized rounding algorithm introduced by Crane et al. (Crane et al., 2025). While the algorithm itself remains unchanged, our refined analysis achieves a significantly better approximation ratio.

### 3.1. The Algorithm

The algorithm, presented as Algorithm 1, operates in two phases. First, for each color $c \in [k]$, a random threshold $\alpha_c \sim \text{Uniform}[0, 1]$ is sampled. A vertex $v$ *tentatively* selects color $c$ if $\alpha_c < x_v^c$. Second, conflicts (vertices with multiple tentative colors) are resolved using a random permutation $\pi$ of colors: each vertex selects the tentative color that appears first in $\pi$.

---

**Algorithm 1** Randomized Rounding for MAXECC

**Input:** LP solution $\{x_v^c\}_{v \in V, c \in [k]}$.
**Output:** Coloring $\lambda : V \to [k]$.
$\pi \leftarrow$ uniform random ordering of colors $[k]$
For $c \in [k]$, $\alpha_c \leftarrow$ uniform random threshold in $[0, 1]$
**for** $v \in V$ **do**
    $\mathcal{W} = \{c \in [k] : \alpha_c < x_v^c\}$
    **if** $|\mathcal{W}| > 0$ **then**
        $\lambda(v) \leftarrow \text{argmax}_{c \in \mathcal{W}} \, \pi(c)$
    **else**
        $\lambda(v) \leftarrow$ arbitrary color
    **end if**
**end for**

---

## 3.2. Analysis

**Theorem 3.1.** *Algorithm 1 is a $\frac{1}{r+1}$-approximation for* MAXECC.

*Remark* 3.2. Crane et al. (Crane et al., 2025) analyzed the same algorithm and obtained ratio $(\frac{2}{e})^r \cdot (r+1)^{-1}$. Their analysis works by bounding the probability of the event that each vertex of an edge tentatively chooses at most one additional color (in addition to the color of the edge).

*Proof.* First observe that the LP optimal is a relaxation of MAXECC, hence greater than the optimal of the problem. We shall compare the expected number of satisfied edges to the LP optimal, and show that $\mathbb{E}[\sum_{e\in E} Z_e] \geq \frac{1}{r+1}\sum_{e\in E} z_e$. By the linearity of expectation, it suffices to show that for any edge $e \in E$, we have $\mathbb{E}[Z_e] \geq \frac{z_e}{r+1}$.

Consider an edge $e$ with color $c_e = \ell(e)$. We denote by $X_v^i$ the indicator that vertex $v$ tentatively selects color $i$ (this occurs independently for each color with probability $x_v^i$), and by $\mathcal{W}_e$ the set of tentative colors (excluding $c_e$) for all vertices in $e$.

Since the LP constraint requires $x_v^{c_e} \geq z_e$ for all $v \in e$, the event $\alpha_{c_e} < z_e$ ensures that all vertices in $e$ tentatively select $c_e$. This event occurs with probability $z_e$. Edge $e$ is satisfied if and only if $\alpha_{c_e} < z_e$ and the permutation $\pi$ ranks $c_e$ before all colors in $\mathcal{W}_e$.

We analyze the probability that $c_e$ comes first conditioned on $\alpha_{c_e} < z_e$:

$$\Pr[c_e \text{ first} \mid \alpha_{c_e} < z_e]$$
$$= \mathbb{E}\left[\frac{1}{1 + |\mathcal{W}_e|} \mid \alpha_{c_e} < z_e\right]$$
$$= \mathbb{E}\left[\frac{1}{1 + \sum_{i\neq c_e} \max_{v\in e} X_v^i}\right]$$
$$\geq \mathbb{E}\left[\frac{1}{1 + \sum_{i\neq c_e} \sum_{v\in e} X_v^i}\right]$$
$$= \mathbb{E}\left[\frac{1}{1 + \sum_{v\in e} \sum_{i\neq c_e} X_v^i}\right]$$
$$\geq \frac{1}{1 + \sum_{v\in e} \sum_{i\neq c_e} \mathbb{E}[X_v^i]}$$
$$= \frac{1}{1 + \sum_{v\in e} \sum_{i\neq c_e} x_v^i}$$
$$\geq \frac{1}{1 + r}$$

The inequality in line 4 uses $\max_{v\in e} x_v^i \leq \sum_{v\in e} x_v^i$. The inequality in line 6 applies Jensen's inequality (since $f(x) = 1/(1+x)$ is convex for $x \geq 0$) and the linearity of expectation. The final inequality uses $\sum_{i\neq c_e} x_v^i \leq 1$ for

---

**Algorithm 2** DEPROUND Subroutine

**Input:** Fractional values $\{(x_v^1, x_v^2)\}_{v\in V}$.
**Output:** Rounded values with $\min(x_v^1, x_v^2) = 0$ for all $v$.
$\gamma \leftarrow \max_v(x_v^1 + x_v^2)$
**while** $\exists v : x_v^1 \neq 0$ and $x_v^2 \neq 0$ **do**
  $\Delta^1 \leftarrow \min(\min_{v:x_v^1\neq 0} x_v^1, \min_{v:x_v^2\neq\gamma}(\gamma - x_v^2))$
  $\Delta^2 \leftarrow \min(\min_{v:x_v^2\neq 0} x_v^2, \min_{v:x_v^1\neq\gamma}(\gamma - x_v^1))$
  $\alpha \leftarrow \text{Uniform}[0, 1]$
  **if** $\alpha < \frac{\Delta^1}{\Delta^1+\Delta^2}$ **then**
    **for** $v : x_v^1 \notin \{0,\gamma\}$ **do**
      $x_v^1 \leftarrow x_v^1 + \Delta^2$
    **end for**
    **for** $v : x_v^2 \notin \{0,\gamma\}$ **do**
      $x_v^2 \leftarrow x_v^2 - \Delta^2$
    **end for**
  **else**
    **for** $v : x_v^1 \notin \{0,\gamma\}$ **do**
      $x_v^1 \leftarrow x_v^1 - \Delta^1$
    **end for**
    **for** $v : x_v^2 \notin \{0,\gamma\}$ **do**
      $x_v^2 \leftarrow x_v^2 + \Delta^1$
    **end for**
  **end if**
**end while**

---

each $v \in e$ (from the LP constraint $\sum_i x_v^i = 1$), giving $\sum_{v\in e} \sum_{i\neq c_e} x_v^i \leq |e| \leq r$.

Therefore,

$$\Pr[Z_e] = \Pr[\alpha_{c_e} < z_e] \cdot \Pr[c_e \text{ first} \mid \alpha_{c_e} < z_e]$$
$$\geq z_e \cdot \frac{1}{r+1}$$

Hence Algorithm 1 achieves expected value at least $\frac{1}{r+1}\sum_{e\in E} z_e \geq \frac{1}{r+1} \cdot \text{OPT}$. We prove the tightness of this approximation in the Appendix. $\square$

# 4. Dependent Rounding: $\frac{1}{\lceil k/2\rceil}$-Approximation

We now present a rounding algorithm based on dependent rounding that achieves approximation ratio $\frac{1}{\lceil k/2\rceil}$ where $k$ is the number of colors. This approach is particularly effective when $k$ is small.

## 4.1. The DEPROUND Subroutine

Our algorithm uses a subroutine based on dependent rounding (Gandhi et al., 2006). Given fractional values $\{(x_v^1, x_v^2)\}_{v\in V}$ for two colors, Algorithm 2 rounds them such that for each vertex $v$, at least one of $x_v^1, x_v^2$ becomes zero.

**Lemma 4.1.** *Algorithm 2 terminates in at most $2|V|$ steps*

*and ensures $x_v^1, x_v^2 \leq \gamma$ for all $v$. Moreover:*

1. *If initially $x_v^i \geq x_u^i$, then after rounding $x_v^i \geq x_u^i$ (order preservation).*

2. *$\mathbb{E}[x_v^i] = x_v^i$ where $x_v^i$ is the initial value (expectation preservation).*

*Proof.* At each iteration, at least one variable reaches 0 or $\gamma$ and stops being updated. Since a variable can only reach $\gamma$ if its paired variable is already 0 (their sum remains constant at $\leq \gamma$ while both are active), the algorithm terminates in at most $2|V|$ steps.

Order preservation holds because all non-boundary values of $x_v^i$ are updated by the same amount in each iteration.

For expectation preservation, in each iteration where $x_v^i$ is updated, it increases by $\Delta^j$ with probability $\frac{\Delta^{-j}}{\Delta^1 + \Delta^2}$ and decreases by $\Delta^{-j}$ with probability $\frac{\Delta^j}{\Delta^1 + \Delta^2}$ (where $\{i, j\} = \{1, 2\}$). The expected change is therefore zero, preserving the expectation. $\square$

### 4.2. Main Algorithm

Algorithm 3 applies DEPROUND in tournament fashion over $\lceil \log k \rceil$ rounds. In each round, we pair up colors and round each pair independently. After rounding, each vertex retains at most one color from each pair. We track which color each vertex retains in a list $P_{v,i}$ for each round $i$.

**Theorem 4.2.** *Algorithm 3 is a $\frac{1}{\lceil k/2 \rceil}$-approximation for* MAXECC.

*Proof.* It suffices to show that for any edge $e \in E$ with color $c_e = \ell(e)$, we have $\mathbb{E}[Z_e] \geq \frac{z_e}{\lceil k/2 \rceil}$.

Let $u = \operatorname{argmin}_{v \in e} x_v^{c_e}$, so $x_u^{c_e} = z_e$ by the LP constraints. By Lemma 4.1, the order is preserved across all rounds, so $x_{v,i}^{c_e} \geq x_{u,i}^{c_e}$ for all $v \in e$ and all rounds $i$. Therefore, if $x_{u,\lceil \log k \rceil}^{c_e} \neq 0$, then $x_{v,\lceil \log k \rceil}^{c_e} \neq 0$ for all $v \in e$, meaning edge $e$ is satisfied. Hence:

$$\mathbb{E}[Z_e] \geq \Pr[x_{u,\lceil \log k \rceil}^{c_e} \neq 0]$$

By Lemma 4.1, $\mathbb{E}[x_{u,\lceil \log k \rceil}^{c_e}] = x_u^{c_e} = z_e$. We now bound the maximum value $x_{u,\lceil \log k \rceil}^{c_e}$ can attain.

After the first round of pairings, for each pair of colors $(2j - 1, 2j)$, by Lemma 4.1:

$$x_{u,2}^{\text{color from pair } j} \leq \max_{v \in V}(x_{v,1}^{2j-1} + x_{v,1}^{2j})$$
$$= \max_{v \in V}(x_v^{2j-1} + x_v^{2j}) \leq 1$$

where the last inequality uses the LP constraint $\sum_c x_v^c = 1$.

**Algorithm 3** Dependent Rounding for MAXECC

**Input:** LP solution $\{x_v^c\}_{v \in V, c \in [k]}$.
**Output:** Coloring $\lambda : V \to [k]$.
Randomly relabel colors to $[k] = \{1, 2, \dots, k\}$
$\forall v \in V, c \in [k]: x_v^c \leftarrow x_v^c$
$\forall v \in V: P_{v,1} \leftarrow [k]$ {List of colors}
**for** $i = 1$ **to** $\lceil \log k \rceil$ **do**
  $\forall v: P_{v,i+1} \leftarrow [\ ]$ {Empty list}
  **for** $j = 1$ **to** $\lceil |P_{v,i}|/2 \rceil$ **step** 2 **do**
    **if** $j = |P_{v,i}|$ **then**
      {Odd number: bye}
      Append $P_{v,i}[j]$ to $P_{v,i+1}$
    **else**
      $\{(x_v^{P_{v,i}[j]}, x_v^{P_{v,i}[j+1]})\}_{v \in V} \leftarrow$
        DEPROUND$(\{(x_v^{P_{v,i}[j]}, x_v^{P_{v,i}[j+1]})\}_{v \in V})$
      **for** $v \in V$ **do**
        **if** $x_v^{P_{v,i}[j]} \neq 0$ **then**
          Append $P_{v,i}[j]$ to $P_{v,i+1}$
        **else**
          Append $P_{v,i}[j + 1]$ to $P_{v,i+1}$
        **end if**
      **end for**
    **end if**
  **end for**
**end for**
$\forall v \in V: \lambda(v) \leftarrow P_{v,\lceil \log k \rceil}[1]$

In subsequent rounds, when combining results from two previous pairs, the new maximum value is at most the sum of the maximum values of the two previous pairs, so:

$$x_{u,\lceil \log k \rceil}^{c_e} \leq \sum_{j=1}^{\lceil k/2 \rceil} \max_{u \in V} x_{u,2}^{\text{color from pair } j}$$
$$\leq \lceil k/2 \rceil$$

Therefore:

$$z_e = \mathbb{E}[x_{u,\lceil \log k \rceil}^{c_e}] \leq \Pr[x_{u,\lceil \log k \rceil}^{c_e} \neq 0] \cdot \lceil k/2 \rceil$$
$$\Pr[Z_e] \geq \Pr[x_{u,\lceil \log k \rceil}^{c_e} \neq 0] \geq \frac{z_e}{\lceil k/2 \rceil}$$

$\square$

*Remark* 4.3. For $k = 2$ (binary edge colors), this gives an exact solution. When $k$ is small relative to $r$, this can outperform the $\frac{1}{r+1}$ and $\frac{1-e^{-r}}{r}$ guarantee of Section 3 and 5.

## 5. Scaling and Refined Analysis

We now introduce a modification to Algorithm 1 that scales the LP values using an exponential function, combined with a refined probabilistic analysis that goes beyond Jensen's inequality.

## 5.1. Modified Algorithm with Scaling

The key idea is to replace the thresholding condition $\alpha_c < x_v^c$ with $\alpha_c < g(x_v^c)$ where $g(x) = x \cdot f(x)$ and $f(x) = e^{-wx}$ for a parameter $w \in (2/3, 1)$ to be chosen.

The modified algorithm assigns color $c$ tentatively to vertex $v$ if $\alpha_c < g(x_v^c) = x_v^c \cdot e^{-wx_v^c}$. Moreover, we assume that $\max_{e \in E} r(1 - z_e) \leq w(k - 1)$, since for small (constant) $k$, our previous algorithm provides a better factor approximation of $\frac{1}{\lceil \frac{k}{2} \rceil}$.

## 5.2. Key Technical Lemma

The core of our improved analysis is the following result, which shows that certain probability functions are minimized when color "wants" are balanced across vertices.

**Theorem 5.1.** *Let $f(x) = e^{-wx}$ with $w > 2/3$ and $g(x) = x \cdot f(x)$. For any $\ell \geq 1$, $t \geq 2$, and $v \geq 0$ with $v \leq t(1 - z_e)$, define:*

$$h_{\ell,v}(m_1, \ldots, m_t) = \sum_{s=0}^{t} \frac{1}{s + \ell} \sum_{\substack{S \subseteq [t] \\ |S| = s}} \prod_{j \in S} p_j \prod_{j' \notin S} (1 - p_{j'})$$

*where $p_i = g(m_i)$ and $\sum_{i=1}^{t} m_i = v$ with $0 \leq m_i \leq 1 - z_e$ for all $i$.*

*Then $h_{\ell,v}(m_1, \ldots, m_t)$ is minimized when $m_1 = \cdots = m_t = v/t$.*

The proof proceeds by induction on $t$, with the base case ($t = 2$) requiring verification that the derivative condition $\frac{\partial h}{\partial m_2} > \frac{\partial h}{\partial m_1}$ holds when $m_2 > m_1$, which reduces to showing $h_{l,v}$ is increasing when perturbed. The full proof is given in the Appendix but we provide the key insight here.

*Proof sketch.* The base case $t = 2$ reduces to showing that $\varphi(x) = \frac{2x}{(1 - wx)(\ell + 2)} - \frac{e^{wx}}{1 - wx}$ is decreasing on $[0, 1]$. Taking derivatives and simplifying shows that $\varphi'(x) < 0$ when $w(2 - wx) \cdot e^{wx} > 2/3$ for all $x \in [0, 1]$. Since $we^{wx}$ is increasing and $(2 - wx) \geq 1$, it suffices to have $w \in (2/3, 1)$.

The inductive step uses the recurrence:

$$h_{\ell,v}(m_1, \ldots, m_t)$$
$$= p_t h_{\ell,v-m_t}(m_1, \ldots, m_{t-1}) +$$
$$(1 - p_t) h_{\ell+1,v-m_t}(m_1, \ldots, m_{t-1})$$

By the inductive hypothesis, both terms on the right are minimized when $m_1 = \cdots = m_{t-1}$, and a symmetric argument shows this must equal $m_t$. $\square$

## 5.3. Approximation Guarantee

**Theorem 5.2.** *The modified algorithm with scaling function $g(x) = xe^{-wx}$ for $w > 2/3$ achieves approximation ratio:*

$$\min_{e \in E} \frac{1 - e^{-r(1 - z_e)}}{r(1 - z_e)} \cdot e^{-wz_e}$$

*In the worst case ($z_e \to 0$ and $k \to \infty$), this approaches $\frac{1 - e^{-r}}{r}$.*

*Proof.* For an edge $e$ with color $c_e$, the probability that all vertices tentatively select $c_e$ is $g(z_e) = z_e e^{-wz_e}$ (since we now threshold at $g(x_v^{c_e})$ instead of $x_v^{c_e}$).

By Theorem 5.1, given an edge $e$ with color $c_e$ such that $c_e$ is tentatively chosen by all $v \in e$ (causing there to be one preexisting tentative color), the probability that $e$ is satisfied in phase 2 is $h_{1,v}(m_1, \ldots, m_t)$ where $t = k - 1$ and $v = r(1 - z_e)$ represents the total "mass" from other colors.

Furthermore, Theorem 5.1 shows this is minimized when $m_1 = \cdots = m_t = v/t = r(1 - z_e)/(k - 1)$.

Setting $p = g(v/t) = \frac{v}{t} e^{-wv/t}$, we can compute a closed form for $h_{1,v}$. Since all $m_i$ are equal, all $p_i = p$:

$$h_{1,v}(m_1, \ldots, m_t)$$
$$= \sum_{s=0}^{t} \frac{1}{s + 1} \binom{t}{s} p^s (1 - p)^{t-s}$$
$$= \frac{1}{t + 1} \sum_{s=0}^{t} \binom{t + 1}{s + 1} p^s (1 - p)^{t-s}$$
$$= \frac{1}{p(t + 1)} \sum_{s=0}^{t} \binom{t + 1}{s + 1} p^{s+1} (1 - p)^{(t+1)-(s+1)}$$
$$= \frac{1}{p(t + 1)} \left[ (p + (1 - p))^{t+1} - (1 - p)^{t+1} \right]$$
$$= \frac{1 - (1 - p)^{t+1}}{p(t + 1)}$$

**Lower bounds on the approximation:** In the appendix, we prove that the approximation factor is minimized as $k \to \infty$ (equivalently, $t \to \infty$), and provide graphical analysis for $z_e \to 0$ as a necessary condition to minimize the approximation for specific values of $w \in (2/3, 1)$ (the analytical proof of which will be in the full paper).

First, observe that

$$\lim_{t \to \infty} p$$
$$= \lim_{t \to \infty} \frac{r(1 - z)}{t} \lim_{t \to \infty} e^{-w\left(\frac{r(1-z)}{t}\right)} :$$
$$\lim_{t \to \infty} \frac{r(1 - z)}{t} = r(1 - z) \lim_{t \to \infty} \frac{1}{t} = r(1 - z) * 0 = 0$$

and

$$\lim_{t\to\infty} e^{-w\left(\frac{r(1-z)}{t}\right)}$$

$$= e^{-w\lim_{t\to\infty}\left(\frac{r(1-z)}{t}\right)}$$

$$= e^{-w*0} = 1$$

$$\implies \lim_{t\to\infty}\frac{r(1-z)}{t}\lim_{t\to\infty} e^{-w\left(\frac{r(1-z)}{t}\right)}$$

$$= \lim_{t\to\infty} p = 0 * 1 = 0.$$

Moreover,

$$\lim_{t\to\infty}\frac{1}{p(t+1)}$$

$$= \lim_{t\to\infty}\frac{\frac{1}{p}}{t+1} :$$

$$\lim_{t\to\infty} t+1 = \infty$$

and

$$\lim_{t\to\infty}\frac{1}{p} = \infty$$

since $p$ is nonnegative and $\lim_{t\to\infty} p = 0$. By L'Hospital's rule,

$$\lim_{t\to\infty}\frac{\frac{1}{p}}{t+1}$$

$$= \lim_{t\to\infty}\frac{\partial}{\partial t}\left(\frac{1}{p}\right)$$

$$= \lim_{t\to\infty}\frac{1}{r(1-z_e)}\left(e^{wr\frac{(1-z_e)}{t}}\right)\left(1-\frac{wr(1-z_e)}{t}\right) :$$

$$\lim_{t\to\infty}\left(e^{wr\frac{(1-z_e)}{t}}\right)$$

$$= e^{\lim_{t\to\infty}\frac{wr(1-z_e)}{t}} = e^0 = 1$$

and

$$\lim_{t\to\infty} 1-\frac{wr(1-z_e)}{t}$$

$$= 1 - \lim_{t\to\infty}\frac{wr(1-z_e)}{t} = 1 - 0 = 1.$$

Thus,

$$\lim_{t\to\infty}\frac{\frac{1}{p}}{t+1}$$

$$= \lim_{t\to\infty}\frac{\partial}{\partial t}\left(\frac{1}{p}\right)$$

$$= \frac{1}{r(1-z_e)} * 1 * 1$$

$$= \frac{1}{r(1-z_e)}.$$

We then calculate $\lim_{t\to\infty} 1-(1-p)^{t+1} = 1-\lim_{t\to\infty}(1-p)^{t+1}$. We see that, denoting $A = \lim_{t\to\infty}(1-p)^{t+1}$,

$$\ln A = \lim_{t\to\infty}\ln((1-p)^{t+1})$$

$$= \lim_{t\to\infty}(t+1)\ln(1-p)$$

$$= \lim_{p\to0}(t+1)\ln(1-p)$$

Notice that $t \to \infty$ if and only if $p \to 0$, and $\lim_{p\to0}\frac{\ln(1-p)}{-p} = \lim_{t\to\infty}\frac{\ln(1-p)}{-p} = 1$.

Therefore,

$$\ln A$$

$$= \lim_{p\to0}(t+1)\ln(1-p)$$

$$= \lim_{p\to0} -p(t+1)\frac{\ln(1-p)}{-p}$$

$$= -\lim_{p\to0}(t+1)p\lim_{p\to0}\frac{\ln(1-p)}{-p}$$

$$= -\lim_{t\to\infty}(t+1)p$$

$$= -\lim_{t\to\infty}(t+1)\frac{r(1-z_e)}{t}e^{-\frac{wr(1-z_e)}{t}}$$

$$= -r(1-z_e)\left(\lim_{t\to\infty}\frac{t+1}{t}\frac{1}{\lim_{t\to\infty} e^{\frac{wr(1-z_e)}{t}}}\right)$$

$$= -r(1-z_e) * 1 * \frac{1}{e^{\lim_{t\to\infty}\frac{wr(1-z_e)}{t}}}$$

$$= -r(1-z_e) * 1 * 1 = -r(1-z_e)$$

giving us that $\lim_{t\to\infty} 1-(1-p)^{t+1} = 1-A = 1-e^{\ln A} = 1 - e^{-r(1-z_e)}$: our approximation factor is $\frac{1}{r(1-z_e)}(1 - e^{-r(1-z_e)})$, since $\lim_{t\to\infty}\frac{1}{p(t+1)} = \frac{1}{r(1-z_e)}$.

As the approximation factor is minimized when $z_e \to 0$, it is therefore

$$\lim_{z_e\to0}\frac{f(z_e)(1-e^{-r(1-z_e)})}{r(1-z_e)}$$

$$= \lim_{z_e\to0}\frac{e^{-wz_e}}{r(1-z_e)}(1-e^{-r(1-z_e)})$$

$$= \frac{e^{-w\lim_{z_e\to0} z_e}}{r(\lim_{z_e\to0} 1-z_e)}(1-e^{-r(1-\lim_{z_e\to0} z_e)})$$

$$= \frac{1}{r}(1-e^{-r}).$$

$\square$

For graphs ($r = 2$), we have $\frac{1-e^{-2}}{2} \approx 0.432$, a substantial improvement over $1/3$ from our refined analysis and $0.245$ from Crane et al.'s hypergraph algorithm, additionally surpassing Crane et al.'s specialized $0.38$ graph algorithm.

## 6. Conclusion and Open Problems

We have presented three approaches to approximating MAX-ECC: (1) an improved analysis achieving $\frac{1}{r+1}$, (2) a dependent rounding algorithm achieving $\frac{1}{\lceil k/2 \rceil}$, and (3) a scaling technique achieving approximately $\frac{1-e^{-r}}{r}$.

Several interesting directions remain:

- **Optimal scaling:** Is $f(x) = e^{-wx}$ the optimal scaling function, or can different choices further improve the ratio?

- **Combining techniques:** Can dependent rounding be combined with scaling to achieve better guarantees when both $k$ and $r$ are small?

- **Hardness:** What are the optimal approximation ratios achievable for MAXECCas a function of $r$ and $k$? Are there improved hardness results?

- **Extensions:** Can these techniques be adapted to overlapping or robust variants of ECC studied by Lee et al. (Lee et al., 2025)?

## Acknowledgments

Jiayi Wu was supported in part by NSF award number CCF-1918749.

## Impact Statement

This paper presents work whose goal is to advance the fields of machine learning and computer science. There are many potential societal consequences of applications of our work, such as those highlighted in the introduction and in the papers that we build on, e.g., (Crane et al., 2025).

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

## A. Full Proof of Theorem 5.1

We restate the theorem for completeness:

**Theorem A.1** (Theorem 5.1 restated). *Let $f(x) = e^{-wx}$ with $w > 2/3$ and $g(x) = x \cdot f(x)$. For any $\ell \geq 1$, $t \geq 2$, and $v \geq 0$ with $v \leq t(1 - z_e)$, define:*

$$h_{\ell,v}(m_1, \ldots, m_t) = \sum_{s=0}^{t} \frac{1}{s + \ell} \sum_{\substack{S \subseteq [t] \\ |S|=s}} \prod_{j \in S} p_j \prod_{j' \notin S} (1 - p_{j'})$$

*where $p_i = g(m_i)$ and $\sum_{i=1}^{t} m_i = v$ with $0 \leq m_i \leq 1 - z_e$ for all $i$.*

*Then $h_{\ell,v}(m_1, \ldots, m_t)$ is minimized when $m_1 = \cdots = m_t = v/t$.*

*Proof.* We prove by induction on $t$ that $\forall t \in \mathbb{Z}^+ \setminus \{1\}$, $\forall \ell \in \mathbb{Z}^+$, $h_{\ell,v}(m_1, \ldots, m_t)$ is minimized when $m_1 = \cdots = m_t = v/t$.

**Base case:** $t = 2$

We start with $t = 2$, and are given that $m_1 + m_2 = v$. In order to show that $\forall \ell \in \mathbb{Z}^+$, $h_{\ell,v}$ is minimized when $m_1 = m_2$ (in which case, $m_1 = m_2 = v/2$), we must show that if without loss of generality $m_2 > m_1$ and $m_2 \to m_2 + \epsilon$ and $m_1 \to m_1 - \epsilon$ for some $\epsilon > 0$ (thus preserving $m_1 + m_2 = v$), then the value of $h_{\ell,v}(m_1, m_2)$ increases.

Note that the change in the value of $h_{\ell,v}$ when $m_2 \to m_2 + \epsilon$ and $m_1 \to m_1 - \epsilon$ is $\epsilon(h_{\ell,v}^{m_2} - h_{\ell,v}^{m_1})$, where $h_{\ell,v}^x$ denotes $\frac{\partial h_{\ell,v}}{\partial x}$.

Observe that

$$h_{\ell,v}(m_1, m_2)$$
$$= (1 - p_1)(1 - p_2)\frac{1}{\ell} + \frac{p_1(1 - p_2) + p_2(1 - p_1)}{\ell + 1} + \frac{p_1 p_2}{\ell + 2}$$
$$= p_1 p_2 \left( \frac{1}{\ell} - \frac{2}{\ell + 1} + \frac{1}{\ell + 2} \right) + (p_1 + p_2) \left( -\frac{1}{\ell} + \frac{1}{\ell + 1} \right) + \frac{1}{\ell}$$
$$= p_1 p_2 \left( \frac{2(\ell + 1)}{\ell(\ell + 2)} - \frac{2}{\ell + 1} \right) - (p_1 + p_2) \left( \frac{1}{\ell(\ell + 1)} \right) + \frac{1}{\ell}$$
$$= \frac{1}{\ell(\ell + 1)} T + \frac{1}{\ell}$$

where

$$T = 2p_1 p_2 \left[ \frac{(\ell + 1)^2 - (\ell^2 + 2\ell)}{\ell + 2} \right] - (p_1 + p_2) = \frac{2p_1 p_2}{\ell + 2} - (p_1 + p_2).$$

For $h_{\ell,v}^{m_2} - h_{\ell,v}^{m_1} > 0$ to hold, we must have $T_2 - T_1 > 0$, where $T_x = \frac{\partial T}{\partial x}$.

Moreover,

$$T_2 = \frac{2p_1}{\ell + 2} g'(m_2) - g'(m_2),$$
$$T_1 = \frac{2p_2}{\ell + 2} g'(m_1) - g'(m_1),$$
$$\Rightarrow T_2 - T_1 = g'(m_2)g'(m_1) \left( \varphi(m_1) - \varphi(m_2) \right)$$

where $\varphi(m_i) = \frac{1}{g'(m_i)} \left( \frac{2p_i}{\ell + 2} - 1 \right)$.

As $g(x)$ is an increasing function on $[0, 1]$, $g'(m_2)g'(m_1) > 0$. Thus, $T_2 - T_1 > 0$ if and only if $\varphi(m_1) - \varphi(m_2) > 0$: this holds if $\varphi(x)$ is a decreasing function on $[0, 1]$.

Letting $f(x) = e^{-wx}$ where $w \in (0,1)$, we have $g(x) = xe^{-wx}$ and $g'(x) = (1 - wx)e^{-wx}$, so:

$$
\begin{aligned}
&\varphi(x) \\
&= \left( \frac{2xe^{-wx}}{\ell + 2} - 1 \right) \cdot \frac{1}{(1 - wx)e^{-wx}} \\
&= \frac{2x}{(1 - wx)(\ell + 2)} - \frac{e^{wx}}{(1 - wx)}.
\end{aligned}
$$

Taking the derivative, we have:

$$
\begin{aligned}
&\varphi'(x) \\
&= \frac{(1 - wx) - x(-w)}{(1 - wx)^2} \cdot \frac{2}{\ell + 2} - \\
&\quad \frac{(1 - wx)we^{wx} - e^{wx}(-w)}{(1 - wx)^2} \\
&= \frac{1}{(1 - wx)^2} \left( \frac{2}{\ell + 2} - we^{wx}(2 - wx) \right).
\end{aligned}
$$

For $\varphi'(x) < 0$ on $[0,1]$ to hold, observe that as $\ell \geq 1$, $\frac{2}{\ell+2} \leq \frac{2}{3}$ and as $w \in [0,1]$ implies $\frac{1}{(1-wx)^2} > 0$, we need $w \in [0,1]$ such that $we^{wx}(2 - wx) > \frac{2}{3}$ for all $x \in [0,1]$: it suffices to have $w \in [0,1]$ such that $we^{wx} > \frac{2}{3}$ for all $x \in [0,1]$, since $w, x \in [0,1]$ makes $2 - wx \geq 2 - 1 \cdot 1 = 1$.

Note that $we^{wx}$ is an increasing function on $[0,1]$, since $\frac{\partial}{\partial x}(we^{wx}) = w^2 e^{wx} > 0$ for all $x \in [0,1]$. Thus, $we^{wx} \geq we^{w \cdot 0} = w$, so it suffices to have $w > \frac{2}{3}$.

As $\varphi(x)$ is decreasing on $[0,1]$, $T_2 - T_1 > 0$, and therefore $h_{\ell,v}^{m_2} - h_{\ell,v}^{m_1} > 0$. In order to minimize $h_{\ell,v}$, we therefore must have $m_1 = m_2$, and as $m_1 + m_2 = v$, this gives us $m_1 = m_2 = v/2$ for $h_{\ell,v}$ to be minimized.

## Inductive Step

We will prove the following recurrence:

$$
\begin{aligned}
&h_{\ell,v}(m_1, \ldots, m_t) \\
&= p_t h_{\ell, v - m_t}(m_1, \ldots, m_{t-1}) \\
&\quad + (1 - p_t) h_{\ell+1, v - m_t}(m_1, \ldots, m_{t-1}).
\end{aligned}
$$

Observe that $h_{\ell,v}(m_1, \ldots, m_t)$ is the probability, conditioned on the fact that outside of colors in $[t]$, there are $\ell$ preexisting tentative colors for the edge $e$, that edge $e$ is satisfied in phase two of our modified algorithm. This gives us that

$$
\begin{aligned}
&p_t h_{\ell, v - m_t}(m_1, \ldots, m_{t-1}) \\
&= \Pr[e \text{ is satisfied} \mid t \text{ is tentative and } |\mathcal{W}_e| \geq \ell]
\end{aligned}
$$

and

$$
\begin{aligned}
&(1 - p_t) h_{\ell+1, v - m_t}(m_1, \ldots, m_{t-1}) \\
&= \Pr[e \text{ is satisfied} \mid t \text{ is not tentative and } |\mathcal{W}_e| \geq \ell]
\end{aligned}
$$

By Bayes's rule, our recurrence holds.

Recall that if $h_{\ell,v}(m_1, \ldots, m_t)$ is minimized when $m_1 = \cdots = m_t$, then as $\sum_{i \in [t]} m_i = v$, we will have $m_1 = \cdots = m_t = v/t$.

In order to show that $h_{\ell,v}(m_1, \ldots, m_t)$ is minimized when $m_1 = \cdots = m_t$, similar to the base case, if given any two variables $m_2$ and $m_1$ with $m_1 < m_2$ such that $m_2 \to m_2 + \epsilon$ and $m_1 \to m_1 - \epsilon$ for some $\epsilon > 0$ (thus preserving $m_1 + m_2 = v$), then the value of $h_{\ell,v}(m_1, m_2, \ldots, m_t)$ increases. We will proceed in cases.

**Case 1:** Suppose that both $m_1, m_2 \in \{m_1, \ldots, m_{t-1}\}$. Here, we will have that

$$h_{\ell,v}(m_1, \ldots, m_t)^{m_2} - h_{\ell,v}(m_1, \ldots, m_t)^{m_1}$$
$$= (1 - p_t)(h_{\ell,v-m_t}(m_1, \ldots, m_{t-1})^{m_2} - h_{\ell,v-m_t}(m_1, \ldots, m_{t-1})^{m_1})$$
$$+ p_t(h_{\ell+1,v-m_t}(m_1, \ldots, m_{t-1})^{m_2} - h_{\ell+1,v-m_t}(m_1, \ldots, m_{t-1})^{m_1}).$$

By the induction hypothesis, both $h_{\ell,v-m_t}(m_1, \ldots, m_{t-1})$ and $h_{\ell+1,v-m_t}(m_1, \ldots, m_{t-1})$ are minimized when $m_1 = \cdots = m_{t-1} = \frac{v-m_t}{t-1}$. Since $m_2 > m_1$, $1 - p_t > 0$, and $p_t > 0$, both terms are positive, thus

$$h_{\ell,v}(m_1, \ldots, m_t)^{m_2} - h_{\ell,v}(m_1, \ldots, m_t)^{m_1} > 0$$

when $m_1, m_2 \in \{m_1, \ldots, m_{t-1}\}$. Thus, to minimize $h_{\ell,v}(m_1, \ldots, m_t)$ we must have $m_1 = \cdots = m_{t-1}$.

**Case 2:** We need to show that minimizing $h_{\ell,v}(m_1, \ldots, m_t)$ requires $m_t = m_j$ for some $j \in [t-1]$, as this would make $m_1 = \cdots = m_{t-1} = m_t = v/t$. Let us choose $j \in [t-1]$ given that one of $m_1, m_2$ is $m_t$ and $m_1, m_2 \neq m_j$.

Denoting $V = \{m_1, \ldots, m_t\} \setminus \{m_j\}$, notice that

$$h_{\ell,v}(m_1, \ldots, m_t)$$
$$= (1 - p_j)h_{\ell,v-m_j}(m_i \mid m_i \in V) + p_j h_{\ell+1,v-m_j}(m_i \mid m_i \in V)$$

since $h_{\ell,v}(m_1, \ldots, m_t)$ is symmetric in the variables $m_1, \ldots, m_t$. Thus following a symmetric argument as above, we see that in order to minimize $h_{\ell,v}(m_1, \ldots, m_t)$, we must have $\forall m_i, m_k \in V, m_i = m_k$. As $\{m_1, \ldots, m_{t-1}\}$ and $V$ have nonempty intersection, we have that in order to minimize $h_{\ell,v}(m_1, \ldots, m_t)$, we must have $m_1 = \cdots = m_t = v/t$. $\square$

## B. Tightness of Approximation Ratios

We now demonstrate that both the $\frac{1}{r+1}$ approximation (Theorem 3.1) and aspects of the scaling-based approximation are tight in the sense that there exist instances where these bounds are achieved.

### B.1. Tightness of $\frac{1}{r+1}$ Initial Approximation

**Proposition B.1.** *There exists a hypergraph instance where the approximation ratio of Algorithm 1 is exactly $\frac{1}{r+1}$.*

*Proof.* Consider a hypergraph $H$ with rank $r$ and $k$ colors where $k = r + 1$. Consider an edge $e$ with color $c_e$ and $|e| = r$ such that in the LP solution, $\max_{v \in e} x_v^{c_e} = \epsilon$ for all $v \in e$ and some small $\epsilon > 0$.

Moreover, configure the LP values for all other colors $c \neq c_e$ and vertices in $e$ as follows: let $e = \{v_1, \ldots, v_r\}$ and let the other colors be $\{c_1, \ldots, c_r\}$ (where we renumber so $c_e$ is excluded). As shown in Table 2.

*Table 2.* LP Values for Tight Instance

|       | $c_e$        | $c_1$        | $c_2$        | $\cdots$ | $c_r$        |
|-------|--------------|--------------|--------------|----------|--------------|
| $v_1$ | $\leq \epsilon$ | $x_{v_1,c_1}$ | $0$          | $\cdots$ | $0$          |
| $v_2$ | $\leq \epsilon$ | $0$          | $x_{v_2,c_2}$ | $\cdots$ | $0$          |
| $\vdots$ | $\leq \epsilon$ | $\vdots$  | $\vdots$     | $\ddots$ | $\vdots$     |
| $v_r$ | $\leq \epsilon$ | $0$          | $0$          | $\cdots$ | $x_{v_r,c_r}$ |

By the LP constraint $\sum_c x_v^c = 1$, we have $\forall i \in [r]$, $x_{v_i,c_i} = 1 - x_{v_i,c_e} \geq 1 - \epsilon \approx 1$.

Since each $v_i \in e$ has exactly one potential tentative color $c_i$ (besides $c_e$) with $x_{v_i,c_i} \geq 1 - \epsilon$, each $v_i$ will tentatively select $c_i$ with probability approximately 1. Thus, with probability at least $(1 - \epsilon)^r \approx 1$, edge $e$ has all $r + 1$ colors tentative (including $c_e$), giving:

$$\Pr[\text{edge } e \text{ satisfied} \mid c_e \text{ tentative}]$$

$$= \Pr[c_e \text{ comes first among tentative colors} \mid c_e \text{ tentative}]$$

$$\approx \frac{1}{r+1}$$

Thus, for this configuration, the approximation ratio is exactly $\frac{1}{r+1}$ (in the limit as $\epsilon \to 0$). □

### B.2. Worst-case for Scaling Algorithm

For the scaling algorithm, we showed that the approximation factor is minimized when $z_e \to 0$ and $k \to \infty$. We now verify this is indeed a worst-case by providing graphical analysis for minimization as $z_e \to 0$ (the analytical proof will be in the full paper) and proving that, given the approximation's minimization as $z_e \to 0$, that the approximation is thus minimized when $k \to \infty$ as well.

**Proposition B.2.** *For the scaling algorithm with $f(x) = e^{-wx}$ and $w > 2/3$, the approximation ratio is minimized when $z_e \to 0$ and $t \to \infty$.*

*Proof.* We start by providing graphs of the approximation factor increasing as a function of $z_e$ on $(0, 1)$ for specific values of w, demonstrating our claim of the approximation's minimization as $z_e \to 0$ for select values of $w$: it suffices to choose any of the listed values of $w$, since the final approximation factor does not include $w$ and these values satisfy both $w \in (\frac{2}{3}, 1)$ and $r < w(k - 1)$, equivalently $r < wt$.

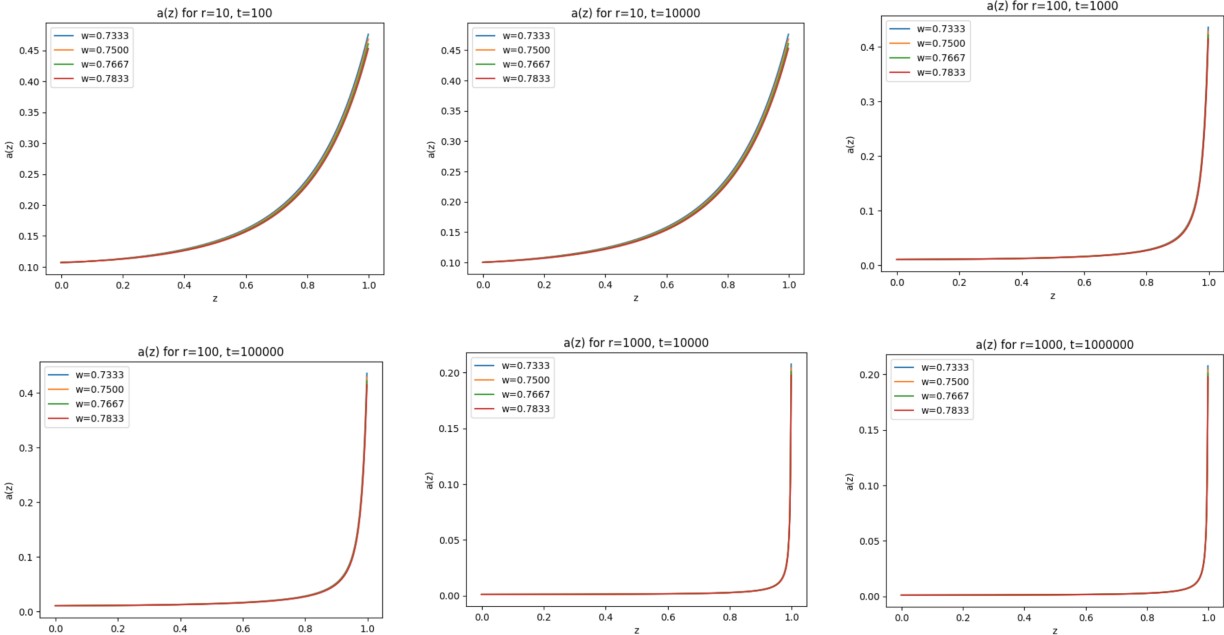

*Figure 1.* Approximation ratio for $z_e \in [0, 1]$

Moreover, we denote $m = \frac{r(1-z_e)}{t}$, where $t = k-1$, making our approximation $f(z_e) \cdot \frac{1-(1-me^{-wm})^{\frac{v}{m}+1}}{(me^{-wm})(\frac{v}{m}+1)}$. As $\lim_{z_e \to 0} m = \lim_{z_e \to 0} \frac{r(1-z_e)}{t} = \frac{r}{t}$, if we can show that our approximation factor is an increasing function of m on $(0, 1)$, thus minimized as $m \to 0$, this proves that our approximation is minimized as $t \to \infty$. It suffices to show that $a(m) = \frac{1-(1-me^{-wm})^{\frac{v}{m}+1}}{(me^{-wm})(\frac{v}{m}+1)}$ is minimized as $m \to 0$, since $f(z_e)$ involves $z_e$ alone.

Taking the derivative of $a(m)$ with respect to $m$ yields

$$\frac{\partial}{\partial m} a(m)$$

$$= \left(me^{-wm}\right) \left(\frac{v}{m} + 1\right) \left(\left(\frac{v}{m} + 1\right)\left(e^{-wm}\right)(1 - mw) \cdot \left(1 - me^{-wm}\right)^{\frac{v}{m}}\right)$$
$$+ \left(\left(1 - me^{-wm}\right)^{\frac{v}{m}+1} - 1\right) \left(e^{-wm}(1 - w(v + m))\right).$$

The first term is always nonnegative, since $mw \leq 1$, $me^{-wm}$ is nonnegative, increasing on $(0, 1)$, achieves a maximum value there of $e^{-w} < 1$, and $\frac{v}{m} + 1 \geq 1$. In order to show that $a(m)$ is increasing on $(0, 1)$, we must show that the second term is nonnegative as well.

As $me^{-wm}$ is increasing and positive on $(0, 1)$, we see that $\left((1 - me^{-wm})^{\frac{v}{m}+1} - 1\right) < (1 - 1)^{\frac{v}{m}+1} - 1) < 0$, requiring us to show that $(e^{-wm}(1 - w(v + m)))$ is negative. This reduces to showing that $1 - w(v + m) < 0$, since $e^{-wm}$ is nonnegative.

We have that $1 - w(v + m) < 0 \iff 1 < w(v + m) \iff 1 < wr(1 - z_e)\frac{t+1}{t}$. Observe that since our approximation factor is minimized when $z_e \to 0$, it suffices to show that $1 < \lim_{z_e \to 0} wr(1 - z_e)\frac{t+1}{t} = \frac{wr(t+1)}{t}$. Since $\frac{t+1}{t} > 1$, we simply need to show that $1 < wr$: this holds because $w \in (\frac{2}{3}, 1)$ and $r \geq 2$: thus, $wr \geq \frac{4}{3} > 1$.

Therefore, our approximation is minimized when both $z_e \to 0$ and $k \to \infty$ (equivalently, $t \to \infty$) occur. $\qquad \square$

