# OpenReview forum: "Edge-colored Clustering in Hypergraphs: A MaxECC Approximation"
_ICML.cc/2026/Conference — ICML 2026 regular_

### Official Review · Reviewer_gVn4 · 2026-03-02

**Soundness:** 3
**Presentation:** 3
**Significance:** 3
**Originality:** 3
**Overall Recommendation:** 4
**Confidence:** 4

**Summary:**

This paper is about edge-colored clustering problem in hypergraphs. Given a hypergraph and an edge coloring (each hyperedge is given one of k colors), the aim is to color the vertices, such that the number of edges that are satisfied by this coloring is maximized. an edge is satisfied by a vertex coloring if all the nodes in this edge receive the same coloring as the edge's color. The algorithms for this problem are all LP based. This paper has three main results:
1) They analyze a previous icml algorithm (Crane et al) and show that the analysis can be significantly improved. This algorithm is based on r the edge size.
2) They give a k/2 approximation algorithm by pairing colors and rounding the coloring on these two colors.
3) They Modify Crane et al rounding algorithm, to boost the approximation. They show that for graphs in particular they improve the approximation to 0.43 from 0.38.

**Compliance With Llm Reviewing Policy:**

Affirmed.

**Final Justification:**

Thanks for the response, will keep my score because of the lack of experiments.

**Key Questions For Authors:**

- can you please expand more on the applicability and importance of this problem? any connections to other problems? any actual helpful usecases in real world that you would want to experiment on?
- page 4, the second equality that replaces $|W_e|$ must be inequality? seems like you are using Jensen's here? if not, can you please explain how you achieve it?

**Limitations:**

discussed in open problems

**Strengths And Weaknesses:**

Overall the paper seems sound. There are a few confusing types (hence why the score of 3 instead of 4). In particular, the definition of $W_e$ is not stated correctly (on page 3), and it seems to be the set of all colors c, such that there is a vertex v in e, where $X_{v,c}$ happens. I point out more in the questions section.

The presentation is overall good.

significance: I am not totally convinced if this problem is being used outside of theory. Or if we are only looking at theoretical applications, it has any connections to any other important problems. Aside from this, they improve past results. Experiments may have favored their manuscript.

originality: They use simple techniques to improve past results which I very much appreciate.

---

> ### Author Rebuttal · Authors · 2026-03-30
>
> We thank the reviewers for their detailed and thoughtful reviews.
> ### Applicability
>
> Clustering is a fundamental problem in ML and data mining, with applications spanning community detection, image segmentation, and collaborative filtering. Edge-colored clustering (ECC) extends traditional clustering to handle **categorical** relationships between data points. Usually, the edges of the hypergraph represent interactions between nodes and the color indicates the category of the interaction.
>
> ECC has been successfully applied to diverse domains. In temporal hypergraph clustering [Amburg et al](https://doi.org/10.1145/3366423.3380152), edge colors encode time windows, enabling the identification of vertices that are especially active during specific periods. In team formation [Amburg et al](https://doi.org/10.1137/1.9781611977172.17), vertices represent individuals, edges correspond to team tasks, and colors indicate task types; ECC then assigns tasks based on prior collaborative experiences. In author-publication networks [Crane et al](https://doi.org/10.48550/arXiv.2502.13000), vertices may be researchers, edges are co-authorship sets, and colors denote publication venues, allowing inference of researchers' primary fields.
>
> In addition to these existing applications, we can outline several real-world problems that ECC models. In **political network analysis**, vertices represent individuals, hyperedges correspond to political forums or online communities (colored by their dominant political affiliation), and MaxECC assigns each individual to a political affiliation that best explains their community memberships; aggregating unsatisfied edges geographically can proxy for political contestedness, offering a data-driven approach to identifying swing states and informing campaign resource allocation. In **industry sector classification**, vertices are companies, hyperedges are joint ventures or supply-chain relationships, and colors are industry sectors; MaxECC seeks a globally consistent sector assignment that maximizes the number of business relationships satisfied within a single sector, naturally handling conglomerates and cross-sector firms as hard cases. In **genomic region classification**, vertices are genomic loci, hyperedges are sets of co-occurring regulatory elements such as enhancer-promoter interactions derived from Hi-C or ChIP-seq data, and colors represent chromatin states or disease-associated annotations (e.g., CpG islands, actively transcribed regions, or loci implicated in a specific disease); MaxECC assigns each locus to a chromatin state or disease association that is maximally consistent with observed regulatory co-occurrence structure, providing a combinatorial approach to genome annotation that respects the multi-way nature of regulatory interactions. Thus, we can use ECC as a framework to classify data objects into a finite number of groups given a classification of their multi-way interactions into finite groups.
>
> MaxECC is also closely related to several well-studied problems. It generalizes max $k$-colored clustering on graphs [Angel et al](https://doi.org/10.1007/978-3-642-40313-2_7), is structurally related to correlation clustering [Bansal et al](doi.org/10.1023/B:MACH.0000033116.57574.95), chromatic correlation clustering [Bonchi et al](doi.org/10.1145/2728170), differing in that edge colors impose categorical constraints rather than similarity weights. The hypergraph setting also connects MaxECC to hypergraph partitioning and community detection problems, where multi-way interactions are essential for capturing higher-order structure that pairwise graph models cannot represent.
>
> ### Explanation of Jensen's inequality
> $W_e$ is indeed the set of all colors $c$ such that there exists a $v\in e$ where $X_{v}^c$ occurs (excluding $l(e)$), which we referred to as the set of tentative colors of vertices in $e$. However, we have made a typo in the equation starting on Line 195: all $x_v^i$ in the equation between Line 200 and Line 210 should have been the uppercase indicator variable $X_v^i$ (so the equality at Line 200 should read $E[\frac{1}{1+ \sum_{i\neq c_e} \max_{v\in e} X_v^i}]$). The inequality on line 209 is the result of Jensen's, and thus the analysis provided remains the same. We apologize for this mistake.

---

> > ### Author Rebuttal · Reviewer_gVn4 · 2026-04-02
> >
> > will keep my score.

---

> > > ### Author Response · Authors · 2026-04-02
> > >
> > > We request the reviewer to see if they would like to increase their score, since we have now shown many applications outside of theory, and also answered their technical questions. Thank you.

---

### Official Review · Reviewer_JtYm · 2026-03-12

**Soundness:** 4
**Presentation:** 4
**Significance:** 3
**Originality:** 4
**Overall Recommendation:** 5
**Confidence:** 4

**Summary:**

The paper studies a variant of edge-colored clustering in hypergraphs called MaxECC. In this problem, we are given a hypergraph $(V, E, \ell)$, where $V$ is the set of vertices, $E$ is the set of hyperedges (each a subset of $V$), and $\ell$ assigns a color to each hyperedge. Each hyperedge also has a non-negative weight. The objective is to color the vertices using the same set of colors as the edges so as to maximize the total weight of satisfied edges, namely those whose vertices all receive the same color as the edge. This problem is NP-hard.

The first contribution of the paper is a tighter analysis of a randomized rounding approximation algorithm proposed for this problem in an ICML 2025 paper. That work introduced the first approximation algorithm for MaxECC, and the authors improve its approximation ratio from $(2/e)^r/(r+1)$ to $1/(r+1)$, where $r$ is the maximum number of vertices in a hyperedge.

Next, the authors explore a different rounding technique, dependent rounding, and obtain an algorithm with approximation ratio $1/\lceil k/2 \rceil$, where $k$ is the number of colors.

Finally, they further improve the approximation ratio of the first algorithm to $(1 - e^{-r})/r$ by modifying the ICML 2025 algorithm through the introduction of a scaling function.

**Compliance With Llm Reviewing Policy:**

Affirmed.

**Final Justification:**

The rebuttal addressed my concerns. My final recommendation is accept.

**Key Questions For Authors:**

I do not have any questions for the authors.

**Limitations:**

yes

**Strengths And Weaknesses:**

**Soundness**

The paper appears technically sound. The claims are supported by rigorous theoretical analysis.

**Presentation**

The paper is well-written and easy to follow. The contributions are clearly stated and are appropriately positioned within the existing literature.

**Significance**

The paper substantially tightens the analysis of the approximation ratio of an existing algorithm, which to my understanding is the only algorithm currently known for this problem, and then further improves it through modifications to the algorithm. It is worth mentioning that the paper also achieves an improved approximation ratio for the special case of graphs, i.e., when $r = 2$. The problem appears relevant to the ML community in practice. However, the absence of experiments somewhat limits the evidence for its practical impact.

**Originality**

The main novelty of the paper lies in the second algorithm, which applies dependent rounding to edge-colored clustering problems. More broadly, the MaxECC problem appears relatively underexplored, and this paper makes meaningful progress toward improving our understanding of it.

---

> ### Author Rebuttal · Authors · 2026-03-30
>
> We thank the reviewers for their detailed and thoughtful reviews.
> ### Experiments
> We have completed tight worst-case analyses for all three of our algorithms, confirming that the approximation ratios we derive are the best our analyses can achieve. We note that in practice, these algorithms are likely to perform significantly better than their worst-case guarantees, as worst-case instances that we discussed are not frequently encountered in real-world hypergraphs. Crane et al. (ICML '25) has verified this empirically for the base randomized rounding algorithm whose analysis we improve; we expect similar, if not stronger, practical behavior for our DepRound and scaling algorithms in their applicable regimes. Should the reviewers deem it necessary, we will include experiments on both synthetic and real-world hypergraphs in the final paper to shed further light on our practical performance.

---

> > ### Author Rebuttal · Reviewer_JtYm · 2026-04-02
> >
> > I believe experiments would be a nice addition to this paper.

---

### Official Review · Reviewer_rtvu · 2026-03-13

**Soundness:** 3
**Presentation:** 3
**Significance:** 3
**Originality:** 3
**Overall Recommendation:** 4
**Confidence:** 4

**Summary:**

This paper studies the MAXECC (Maximum Edge-Colored Clustering) problem on hypergraphs, where the goal is to color vertices to maximize the number of satisfied edges, edges whose vertices all receive the edge's designated color. The paper studies a central concept in clustering with categorical relationships. The authors analyze a notable context of LP-rounding approximation algorithms for this problem. The paper presents three main contributions: (1) an improved analysis of an existing randomized rounding algorithm from Crane et al. (ICML 2025), improving the approximation ratio from $(2/e)^r · 1/(r+1)$ to $1/(r+1).$ (2) a novel dependent-rounding algorithm achieving $2/k$ approximation where k is the number of colors. And (3) a scaling technique with exponential thresholding that achieves $(1−e^{−r})/r$ approximation, notably improving the graph case $(r=2)$ from 0.38 to approximately 0.432.

**Compliance With Llm Reviewing Policy:**

Affirmed.

**Final Justification:**

Thanks for your response.
Will proceed with my initial evaluation.

**Key Questions For Authors:**

Can you clarify the valid parameter regime for graph r =2?

**Limitations:**

The authors acknowledge the lack of empirical evaluation. The theoretical limitations (parameter constraints, incomplete analytical proofs) are not explicitly discussed as limitations but should be.

**Strengths And Weaknesses:**

Strength:
The scaling technique in Section 5 is creative. The key insight that probability functions $h_{l,v}$ are minimized when color assignments are balanced (Theorem 5.1) is non-trivial and requires careful analysis beyond standard Jensen's inequality arguments.
The three techniques address different parameter regimes, the dependent rounding approach is effective when k is small relative to r, while the scaling approach excels for general hypergraphs. This provides practitioners with algorithmic choices based on their problem structure.
Clear improvement over prior art: Table 1 effectively demonstrates consistent improvements across all hypergraph ranks, with particularly notable gains for the practically important graph case (r=2).

Weaknesses:
 While the contributions are theoretical, even simple experiments on synthetic or real-world hypergraphs would strengthen the paper's practical relevance, especially given the clustering applications mentioned in the introduction.

The paper focuses on approximation ratios but does not analyze the running time of the algorithms in detail. The dependent rounding algorithm (Algorithm 3) involves O(log k) rounds with DEPROUND subroutines, what is the overall complexity?

---

> ### Author Rebuttal · Authors · 2026-03-30
>
> We thank the reviewers for their detailed and thoughtful reviews.
>
> ### Complexity
> We have stated the complexity for DepRound to take at most $O(n)=O(|V|)$ "rounds". With proper implementation, i.e., sorting the floating variables and only updating the ends of the lists (handling each "round" in constant time), the runtime of DepRound is $O(n\log n)$. Since each DepRound eliminates at least one color from every vertex, we would require $O(k)$ runs of DepRound, totaling a runtime of $O(kn\log n)$. Considering that the algorithm involves solving an LP with at least $kn$ variables, rounding did not seem like a bottleneck for runtime. We will include this full analysis in the final paper.
>
> ### Parameter regime for r = 2
> The scaling algorithm requires the parameter constraint $r < w(k-1)$ with $w \in (2/3, 1)$. For graphs ($r = 2$), this requires $k > 1 + 2/w$; since $w > 2/3$, the constraint is satisfiable for all $w \in (2/3, 1)$ when $k \geq 5$. For smaller $k$ with $r = 2$, our dependent rounding algorithm (Section 4)
> provides a superior approximation guarantee of $\frac{1}{\lceil k/2 \rceil}$: for example, for $k = 2$ this gives an exact solution, and for $k = 3, 4$ it gives a $1/2$ approximation, both exceeding the scaling algorithm's $\frac{1-e^{-2}}{2} \approx 0.432$ guarantee. The two algorithms are thus complementary, covering different regimes of $k$ relative to $r$.
>
> ### Completeness of Proofs
> We provide a graphical analysis for why the approximation ratio of the scaling technique is minimized as $z_e \to 0$ (a step in showing that the given analysis of the scaling algorithm is tight): we will include the full proof in the final paper. Other than this, the only incomplete proof is that the approximation ratio given by the scaling technique is minimized as $k \to \infty$: even here, outside of the fact that it relies on the approximation's minimization as $z_e \to 0$, the proof is coherent. All other main theorems have complete proofs. Specifically, we show that Crane et al.'s hypergraph coloring algorithm always achieves an approximation factor of at least $\frac{1}{r+1}$ (as well as an instance where this is tight), our dependent rounding algorithm achieves at least a $\frac{1}{\lceil\frac{k}{2}\rceil}$ approximation (where $k$ is the number of colors), and our modified scaling rounding algorithm achieves an $\frac{1 - e^{-r}}{r}$ approximation. What remains to show analytically is that the last approximation is tight when $z_e \to 0$: augmenting the current graphical analysis, we will add a full proof in the final paper.
>
> ### Experiments
> We have completed tight worst-case analyses for all three of our algorithms, confirming that the approximation ratios we derive are the best our analyses can achieve. We note that in practice, these algorithms are likely to perform significantly better than their worst-case guarantees, as worst-case instances that we discussed are not frequently encountered in real-world hypergraphs. Crane et al. (ICML '25) have verified this empirically for the base randomized rounding algorithm whose analysis we improve; we expect similar, if not stronger, practical behavior for our DepRound and scaling algorithms in their applicable regimes. Should the reviewers deem it necessary, we will include experiments on both synthetic and real-world hypergraphs in the final paper to shed further light on our  practical performance.

---

> > ### Author Rebuttal · Reviewer_rtvu · 2026-04-02
> >
> > resolved my questions

---

> > > ### Author Response · Authors · 2026-04-02
> > >
> > > We request the reviewer to see if they would like to increase their score. We have rigorously shown that our algorithms improve upon the previous-best algorithms (and since we have done worst-case analysis, our algorithms can only do better in practice), and also answered the reviewer's technical questions. Please also see our responses to the other reviewers for additional applications.
> > >
> > > We can guarantee to include experimental results if the paper gets accepted, in case the reviewers would like this.
> > >
> > > Thank you.

---

### Official Review · Reviewer_3HNv · 2026-03-14

**Soundness:** 4
**Presentation:** 4
**Significance:** 3
**Originality:** 4
**Overall Recommendation:** 5
**Confidence:** 4

**Summary:**

This paper studies the MAXECC problem. Previous approximation algorithms (like the recent one by Crane et al., 2025) had approximation ratios that decayed exponentially with the hyperedge rank $r$. The authors improve upon this result and present three main theoretical contributions: 1) a refined analysis of the existing randomized rounding algorithm from Crane et al., improving the ratio from an exponential decay to $1/1+r$. a novel dependent-rounding algorithm that gets a $1/⌈k/2⌉$ approximation; and 3) a new scaling technique modifying the thresholding step, which further improves the bound to $(1-e^{-r})/r$​. This scaling technique pushes the state-of-the-art approximation ratio for standard graph MAXECC ($r=2$) from 0.38 to 0.432.

**Compliance With Llm Reviewing Policy:**

Affirmed.

**Final Justification:**

I keep my original very positive review of this paper. The lack of experiments does prevent me from raising the paper to a 6.

**Key Questions For Authors:**

No questions

**Limitations:**

yes

**Strengths And Weaknesses:**

# Strengths

Overall in my opinion this is a strong theoretical result.

**S1** there is clear improvement over state of the art of Crane et al. (2025).

**S2**  this is achieved with several novel techniques. The introduction of dependent-rounding techniques to the ECC literature and the use of balanced "mass" analysis (Theorem 5.1) are novel as far as I'm aware.

**S3** the results seem mathematically sound to me as well. I must also say that the manuscript is very well written - apart from some minor issues that I list below.

# Weaknesses
**W1** the only weakness in this paper in my opinion is the lack of empirical results.  Previous work by Crane et al. (2025) does contain some experimental results to verify the results in practice. I also realise that this is mostly a theoretical paper, so that's why I only consider this a minor weakness. I'd expect that LP-based methods probably perform better (even though they provide worse approx. guarantees).

# minor issues
- hyperdge line 052 should be hyperedge
- large white space after table 1, formatting can be better here
- there is a long dash on line 87 second column
- throughout the manuscript you could consider using \citet instead of \cite or \citep, so the references flow a bit better (instead of having things like Crane et al. (Crane et al., 2025)).

---

> ### Author Rebuttal · Authors · 2026-03-30
>
> We thank the reviewers for their detailed and thoughtful reviews.  We apologize for typos and formatting issues and will fix them promptly.
> ### Experiments
> We have completed tight worst-case analyses for all three of our algorithms, confirming that the approximation ratios we derive are the best our analyses can achieve. We note that in practice, these algorithms are likely to perform significantly better than their worst-case guarantees, as worst-case instances that we discussed are not frequently encountered in real-world hypergraphs. Crane et al. (ICML '25) have verified this empirically for the base randomized rounding algorithm whose analysis we improve; we expect similar, if not stronger, practical behavior for our DepRound and scaling algorithms in their applicable regimes. Should the reviewers deem it necessary, we will include experiments on both synthetic and real-world hypergraphs in the final paper to shed further light on our  practical performance.

---

> > ### Author Rebuttal · Reviewer_3HNv · 2026-04-03
> >
> > Thank you for your response, I keep my original positive score

---

### Decision · Program_Chairs · 2026-04-30

**Decision:**

Accept (regular)

**Comment:**

All reviewers liked the paper, appreciating the strong technical contribution and novel techniques, as well as good writing. The sentiment would be even more positive if an experimental evaluation was added. But without it, it is still a clear accept.